# Binder and Mixture Fatigue Performance of Plant-Produced Road Surface Course Asphalt Mixtures with High Contents of Reclaimed Asphalt

**Ayad Subhy [1],\*, Gustavo Menegusso Pires [2], Ana Jiménez del Barco Carrión [2] , Davide Lo Presti [2,3] and Gordon Airey [2]**

[1] Pavement Design and Asset Management, Civil and Infrastructure, AECOM, Nottingham NG9 6RZ, UK
[2] Nottingham Transportation Engineering Centre, University of Nottingham, Nottingham NG7 2RD, UK
[3] Dipartimento di Ingegneria, Scuola Politecnica, Edificio 8, Universitá degli Studi di Palermo, 90128 Palermo, Italy

\* Correspondence: ayad.subhy@aecom.com; Tel.: +44-115-907-7000

**Abstract:** The aged properties of Reclaimed Asphalt (RA) binders are one of the main factors working against their utilisation in high-RA content (>30%) mixes for surface courses. Fatigue cracking is the main distress of surface courses that are manufactured with a high percentage of RA. This investigation presents results of the rheological and fatigue results of different asphalt mixtures and their recovered binders. The binders were recovered from asphalt mixtures that had been manufactured in asphalt plants using different amounts of RA with contents up to 60% with and without rejuvenators. Two different sources of RA were used, representing a moderately aged RA and an extremely aged RA. The Dynamic Shear Rheometer (DSR) was used to assess the fatigue-characteristics of the binders using time sweep tests while the fatigue characteristics of their mixtures were assessed using the Indirect Tensile Fatigue Test (ITFT). The fatigue data was analysed based on the cumulative dissipated energy approach in addition to traditional fatigue analysis. Results have shown that the ageing condition of RA significantly affects the fatigue properties of recovered binders. Binder and asphalt mixture fatigue results showed that RA contents up to 60% can produce comparable fatigue performance compared to lower percentages of RA in road surface course if the aged RA binder is sufficiently rejuvenated.

**Keywords:** road surface courses; reclaimed asphalt (RA); asphalt binder; fatigue performance; rejuvenators

## 1. Introduction

One aggravating factor for highways is the high consumption of virgin materials required for new construction and maintenance of pavements. Virgin materials associated with flexible pavements are mainly mineral aggregates and bitumen. For aggregates, although material sources are extensive, they are finite and in certain cases not conveniently located. Therefore, the ideal sustainable solution in road engineering is to deal with asphalt pavements as a product with no waste and built with a minimization of virgin material and transport distances. In this sense, recycling pavement methods have gained popularity with pavement and highway management authorities and are seen as politically correct and economically viable alternatives to traditional techniques. The economic and environmental returns that are obtained from recycling asphalt pavement have encouraged more research in this area. Minimizing the virgin materials used in asphalt pavements, e.g. aggregates and bituminous binders, results in economic savings and environmental benefits. Recycling methods are therefore no longer simply considered a green construction solution and have increasingly been adopted across Europe.

Despite the economic and environmental benefits, incorporating high amounts of Recycled Asphalt (RA) in surface layers is limited and not encouraged in Europe. The main concern with this is that the performance requirements of wearing courses can be compromised when the larger volumetric component is replaced by recycled materials. This is due to inherent inconsistency of the RA properties that vary with the milling technique, the maintenance history and in-service life of asphalt pavements. Using high content (>30%) of RA in new asphalt mixtures saves money and natural resources. However, the incorporation of high content RA in pavement surface layers is still limited due to agency specifications (a maximum of about 25%) based on both technical concerns of final mixture performance as well as concerns associated with limitations of existing asphalt plants. Despite these limitations, researchers, highway authorities and the asphalt paving industry are collaborating to investigate and develop innovative techniques and designs that enable higher amounts of RA to be incorporated without compromising pavement performance [1–3]. The use of high RA content in asphalt mixtures has gleaned satisfactory results, proving the potential of using high RA content if it is carefully controlled and monitored [4–10]. Other studies have focused on proposing innovative methods for characterizing reclaimed asphalt and adapting asphalt mixture design methodologies [11–19] to take advantage of the RA component.

Fatigue cracking is the main distress found in surface courses manufactured with a high percentage of RA due to the normally brittle behaviour of RA coupled with repetitive traffic loading at intermediate temperatures. It is known that the fatigue properties of asphalt mixtures are related to the behaviour of their binders and cracking generally starts and propagates within the binder or at the interface boundary between the aggregate and bitumen. Thus, it is appropriate to investigate the performance of bituminous binders and relate their properties to asphalt mixtures and pavement performance. However, it is necessary to characterise the binders based on test procedures and parameters that can assess the binder contribution to fatigue damage resistance correctly. Utilising the Dynamic Shear Rheometer (DSR), the time sweep repeated cyclic loading test allows for the fatigue characteristics to be measured, simulated and monitored directly through the damage behaviour of binders. This method defines the fatigue properties of binders and can be correlated with asphalt mixtures and long-term, field pavement performance [20,21]. Different approaches have been applied to evaluate the fatigue binder damage properties including the dissipated energy approach, which can provide a fundamental fatigue law for bituminous materials [19–25].

This paper presents the rheological properties and fatigue performance-related characteristics of recovered binders from asphalt mixtures manufactured in asphalt plants using different sources and amounts of RA (up to 60%) with and without rejuvenators. The binder fatigue results are then compared to the fatigue properties of the associated asphalt mixtures measured using an indirect tensile testing configuration. These results are essential to provide information about the characteristics of the recycled asphalt binder and the effect of the interaction of additional components such as virgin asphalts and/or rejuvenating additives.

## 2. Materials and Methods

### 2.1. RA and Mixtures

Two RA sources with different degrees of stiffness were studied to address the effect of RA properties on the performance-related properties of asphalt mixtures incorporating RA. RA1 was a moderately stiff surfacing material with a relatively soft recovered binder (penetration >20 dmm) while RA2 was severely stiff with a harder recovered binder (penetration <10 dmm). The binder content of RA1 and RA2 was 4.8% and 5.8%, respectively.

The mixtures selected for the investigation were typical asphalt mixtures used as wearing courses on high traffic volume roads in Germany and Italy and all the mixtures were produced in asphalt plants.

The mixture design of the German asphalt mixtures consisted of stone mastic asphalt (SMA) with a nominal aggregate size of 8 mm (SMA 8 S) and a polymer-modified bitumen (PmB 25/55-55).

The mixture design of the Italian asphalt mixtures consisted of asphalt concrete (AC) with a nominal aggregate size of 16 mm and a tradition penetration grade bitumen. The percentage of RA within each mixture varied from 0% (control mixture) to 60% RA. The comparison between the mechanical properties of the control mixture and the mixtures with increasing content of RA were used to access the influence of RA on mixture performance.

A binder blend design was undertaken using both the conventional binder properties of the binders (recovered RA binder and new virgin binder), as well as the Superpave critical service temperatures for the two case studies [26]. The design showed that the German mixtures did not require rejuvenators, while rejuvenation was required for the Italian mixtures. According to these designs, the various asphalt mixtures were manufactured, and their binders recovered. As each binder blend was designed with different proportions of rejuvenator and virgin binder, it was expected that each of them would have shown similar properties, regardless of the varying amount of RA [27].

The RA binders to be tested were recovered from the asphalt mixtures following the EN 12697-4:2015, Fractionating Column by distillation procedure. It can be observed from Table 1 (binder conventional properties) that the residue binder of RA1 is a softer material, having a penetration value of over 20 and physical properties comparable to the virgin binder. On the contrary, the residue binder of RA2 was very aged with a very low penetration, higher softening point, viscosity and Fraass breaking point. In order to achieve high RA contents in the mixtures, the use of a Rejuvenator (Rej) was required for the Italian asphalt mixtures.

**Table 1.** The conventional properties of binders.

| Binder | Penetration 25 °C (dmm) | Softening Point (°C) | Fraass B. Point (°C) | Viscosity 135 °C (Pa.s) |
|---|---|---|---|---|
| | EN 1426 | EN 1427 | EN 12593 | EN 13302 |
| Recovered binder RA1 | 21.7 | 65.7 | −8 | 1.52 |
| Recovered binder RA2 | 8.3 | 71.4 | +9 | 1.83 |
| PMB used in SMA | 43 | 60.4 | −16 | 1.20 |
| Pen 50/70 used in AC | 68 | 47.6 | −8 | 0.27 |

The rejuvenator used in this study is a combination of a special regenerated oil and wax. This product was chosen because the wax would allow the manufacturing temperature to be reduced while the oil would help rejuvenate the RA binder. The proportion of Rej added to the mixtures was defined by the ratio in weight of Rej/RA-binder being equal to 0.2, where the RA-binder is the effective binder content in each mixture produced and mixed following the supplier recommendations to obtain the adequate effect.

Five different recycled mixtures, in addition to two control virgin mixtures, were produced to represent different cases to be evaluated for fatigue. The rheological and fatigue-related properties tests were conducted on the recovered binders and mixtures shown in Table 2 with the description of each case of study.

**Table 2.** The description of each mixture and binder considered in this study for performance-related characterisations.

| No | Mix. Label | Binder Label | Description |
|----|-----------|-------------|-------------|
| 1 | RAao | Rao | A virgin asphalt mixture (SMA) manufactured using polymer modified bitumen |
| 2 | RAa1 | Ra1 | An asphalt mixture produced using 30% of Reclaimed Asphalt (RA1) and 70% of virgin asphalt mixture (SMA) manufactured using polymer modified bitumen |
| 3 | RAa2 | Ra2 | An asphalt mixture produced using 60% of Reclaimed Asphalt (RA1) and 40% of virgin asphalt mixture (SMA) manufactured using a polymer modified bitumen without rejuvenator |
| 4 | RAa3 | Ra3 | An asphalt mixture produced using 60% Reclaimed Asphalt (RA1) and 40% of virgin asphalt mixture (SMA) manufactured using a polymer modified bitumen plus rejuvenator (combination of a special regenerated oil and wax) |
| 5 | RAbo | Rbo | A virgin asphalt mixture (dense graded) manufactured using a neat paving grade bitumen Pen 50/70 |
| 6 | RAb1 | Rb1 | An asphalt mixture produced using 30% of extremely aged Reclaimed Asphalt (RA2) and 70% of virgin asphalt mixture (dense graded) manufactured using a neat paving grade bitumen PEN 50/70 plus rejuvenator (combination of a special regenerated oil and wax) |
| 7 | RAb2 | Rb2 | An asphalt mixture produced using 60% of extremely aged Reclaimed Asphalt (RA2) and 40% of virgin asphalt mixture (dense graded) manufactured using a neat paving grade bitumen Pen 50/70 plus rejuvenator (combination of a special regenerated oil and wax) |

### 2.1.1. SMA Mixtures

The final gradations of the studied SMA mixtures (all produced in plant) are shown in Figure 1 and conform to the German standard for grading curves "TL-Asphalt StB 07". Differences in final gradation can be related to the assumed inhomogeneity of the RA. The heterogeneity of RA due to various factors including source, milling and crushing activities adds to the difficulty of designing to the standard asphalt mixture gradation. Appropriate characterisation of the RA can result in a reduction of the potential for errors, although they cannot be erased completely.

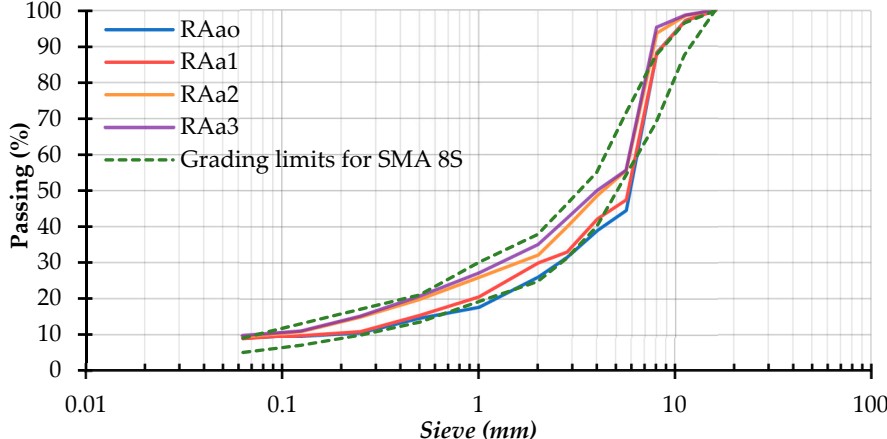

**Figure 1.** Grading limits for SMA 8S and gradation of the tested RA1 asphalt mixtures mixed in plant.

The binder content of the plant-produced mixtures were between 7% and 7.5% and the air voids were equal to 3% for the manufactured testing samples. The maximum temperature during the plant mixing process was defined as 170 °C, with the loose mixed material then shipped to the laboratory and re-heated to 160 °C for about 2 h to produce the test samples with 150 °C being defined as the temperature for compaction.

### 2.1.2. AC Mixtures

The complete gradations of the tested AC mixtures (produced in plant) are shown in Figure 2, in accordance with the Italian standard "Capitolato speciale d'appalto—Norme Tecniche Azienda Nazionale Autonoma delle Strade". Differences in gradation are due to the assumed inhomogeneity of the RA. The RA used for the production of the AC mixtures were considered to be relatively old and required a rejuvenating additive to be added to all the AC mixtures incorporating RA.

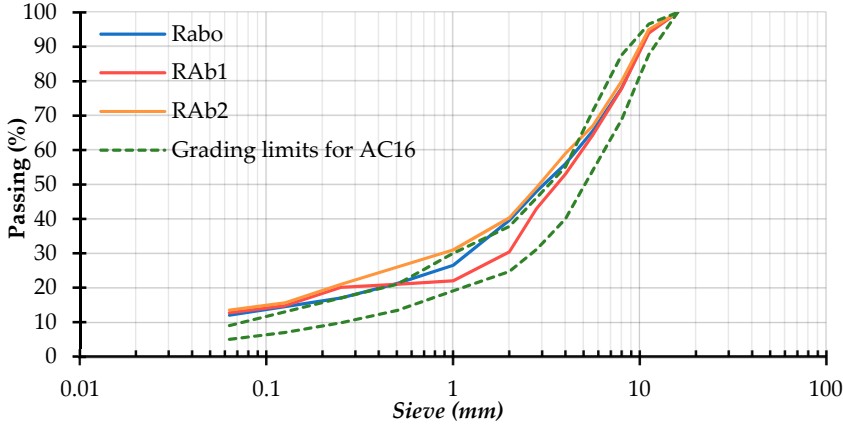

**Figure 2.** Grading limits for AC 16 according to the Italian specification ANAS and gradation of the tested RAP2 asphalt mixtures.

The binder content of the plant-produced mixtures were between 4.5% and 6% and the air voids equal to 3% for the manufactured testing samples. A temperature range during the plant mixing process was defined between 160 and 170 °C. The mixtures shipped to the laboratory were then re-heated to 160 °C for about 2 h in order to produce the test samples with 150–160 °C being defined as the temperature for compaction.

*2.2. Experimental Design*

2.2.1. Binders

Frequency sweep and time sweep tests were undertaken by means of a Dynamic Shear Rheometer (DSR). The NCHRP Report 459 [17] suggested a method to analyse the fatigue performance of binders through the application of a cyclic shear load to sample at a fixed frequency and strain/stress level until failure. The fatigue tests using the DSR were carried out under controlled stress conditions at 20 °C and a frequency of 10 Hz using the 8 mm plate testing geometry with a 2 mm gap. Various stress levels (200 kPa–600 kPa) were applied to the different binders to produce different initial strains. The Dissipated Energy Ratio (DER) concept proposed by Bahia et al. [20] was used to produce an analytical criterion to define the fatigue failure of bituminous materials.

$$DER = \frac{\sum_{i=1}^{n} W_i}{W_n} \qquad (1)$$

where $W_i$ = dissipated energy per cycle, $W_n$ = dissipated energy at cycle $n$. Fatigue laws were then obtained following Equation (2):

$$N_f = a\,(\varepsilon_o)^b \qquad (2)$$

where $N_f$ is the cycles of failure, $\varepsilon_o$ is the initial strain in the test at the set stress level, and *a* and *b* are fitted parameters. Figure 3 shows the relationship between the DER and loading cycles. In the initial portion of the relationship there is only marginal damage in materials with DER = *n*, i.e., the dissipated energy is basically equal between the consecutive loading cycles. As the relative difference in the dissipated energy parameter between consecutive loading cycles becomes significant, the DER starts to deviate from the equality line, which can be interpreted in the test as crack initiation. In this study, the fatigue failure ($N_f$) point can be defined by this rapid change in DER, with this change being considered highly material-specific and independent of the mode of loading [20,28].

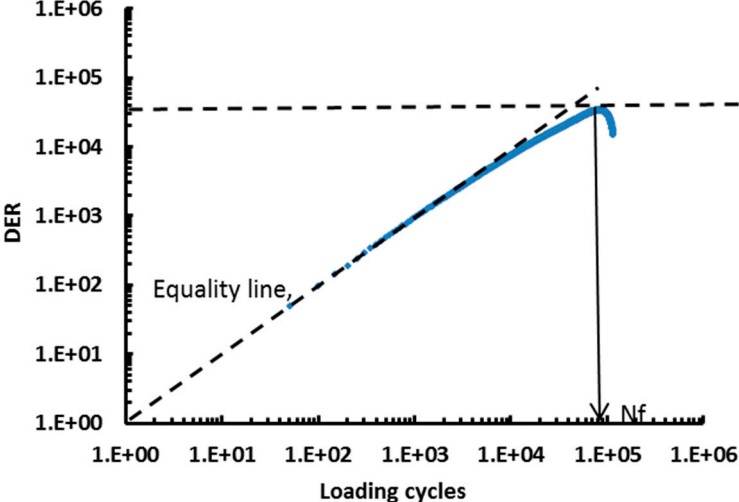

**Figure 3.** Identify the (*Nf*) from the DER vs number of load cycles curve.

The frequency sweep tests were conducted on the binders using the 25 mm parallel plate testing geometry at temperatures from 20 to 80 °C at 10 °C interval and frequencies from 0.1 Hz to 10 Hz. The strain load applied during the frequency sweep tests was 1%, which falls within the linear viscoelastic range of the binders previously determined.

### 2.2.2. Asphalt Mixtures

The Indirect Tensile Fatigue Test (ITFT) on cylindrical specimens is a relatively simple, rapid and cost-effective test method following the European Standard EN 12697-24:2012. Within this paper, the ITFTs was conducted on all the German and Italian asphalt mixes to examine the fatigue-related performance of asphalt mixtures with varying amount of RA. The conducted ITFTs were performed with a constant stress amplitude. The specimens (100 mm diameter by 40 mm height) were subjected to a force controlled harmonic sinusoidal loading with a defined lower and upper level of strain without rest periods. During testing, the specimen was loaded by two diametrically applied compressive forces imparted by means of curved load distribution bars. The resultant horizontal deformations were measured using two Linear Variable Differential Transformers (LVDTs). The conducted indirect tensile fatigue tests were performed at a temperature of 20 °C and a frequency of 10 Hz until specimen failure. The horizontal strains in the centre of the specimen were in a range of 50 to 300 μm/m.

The failure cycle $N_f$/50 was determined when the stiffness of a specimen during testing decreased to 50% of its initial value. To establish the fatigue laws, at least nine samples were tested for each asphalt mixture with the fatigue curves being determined using Equation (3):

$$N_f = k_1 (\gamma_{initial})^{k_2} \tag{3}$$

where:

- Yinitial is the initial elastic strain (%);
- $N_f$ is the number of load cycles until macro-crack;
- $k_1$ and $k_2$ are material parameters.

Once the number of load cycles to failure was determined for each specimen, the number of cycles to failure was plotted versus the corresponding initial elastic horizontal strains on a log-log graph. Then a linear regression equation was fitted to the data points separately for each asphalt mixture. According to the European Standard EN 12697-24:2012, the corresponding coefficient of determination $R^2$ should be >0.9 for asphalt surface layers.

## 3. Results and Discussion

### 3.1. Asphalt Binders

### 3.1.1. Dynamic Mechanical Analysis (DMA)

Evaluation of the viscoelastic rheological properties of binders under different loading times and temperatures allows for performance-related properties of materials at a fundamental level to be determined. This was achieved by means of DMA which was used to determine the stress-strain-time-temperature response of the binders. The rheological data was used to produce master curves of complex modulus |G*| at a reference temperature of 20 °C using the Time-Temperature Superposition Principle (TTSP) and William, Landel and Ferry (WLF) equation, thanks to the thermorheological simplicity of all the binders as shown in Figures 4 and 5 with the Black Diagram of the binders recovered from both mixtures. Figure 6 shows the master curves of recovered binders from SMA mixtures while Figure 7 shows the data of recovered binders from AC mixtures. The master curves show that as the RA content increases, the complex modulus also increases over the frequency domain. The increase at low frequencies (equivalent to high-temperature response) of the complex modulus is considered desirable for the rutting related performance. However, the increase at high frequencies

(equivalent to low-temperature response) could negatively affect the fatigue-related properties. It can be seen that the addition of a rejuvenator has significantly contributed to restoring the flexibility of Ra binders, especially in the case of Ra3 where the complex modulus at high frequencies is even lower than Rao (contains no RA).

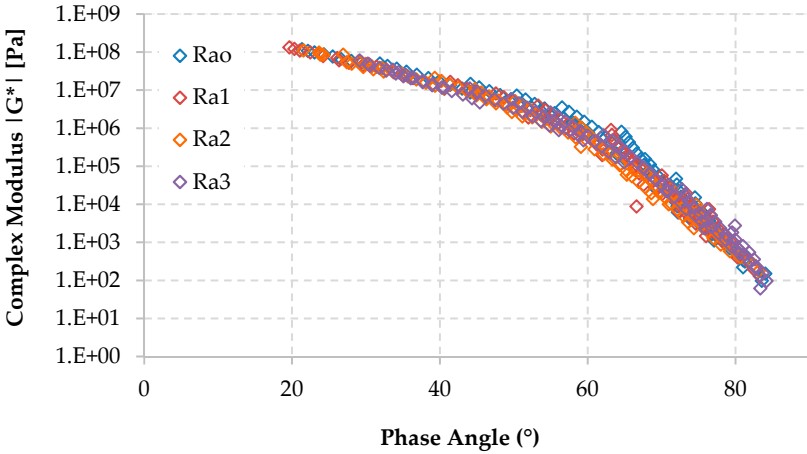

**Figure 4.** Black Diagram for binders recovered from SMA.

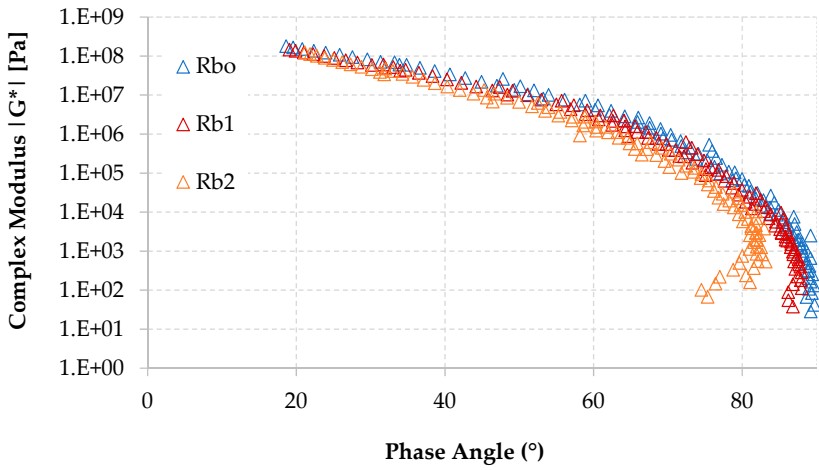

**Figure 5.** Black Diagram for binders recovered from AC.

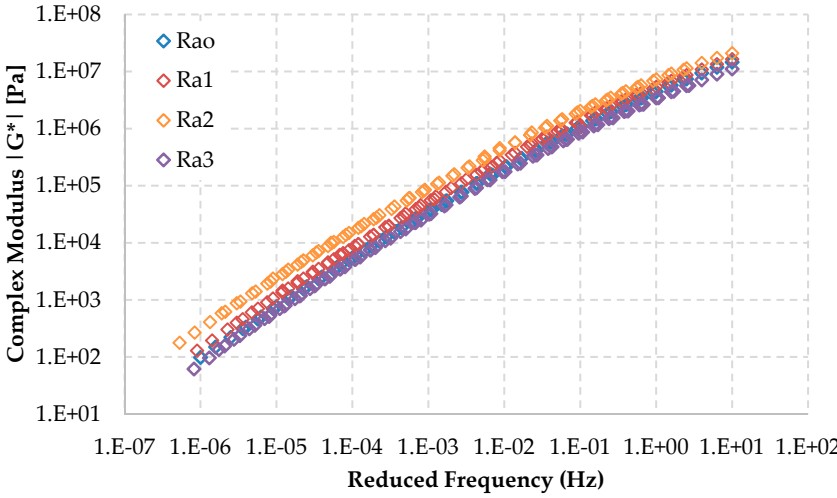

**Figure 6.** Master curve at 20 °C reference temperature for binders recovered from SMA.

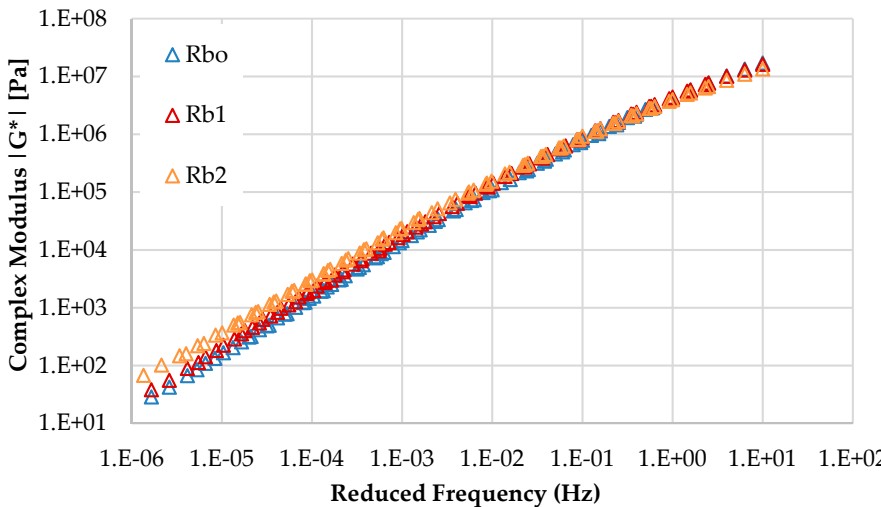

**Figure 7.** Master curve at 20 °C reference temperature for binders recovered from AC.

### 3.1.2. Fatigue Related Analysis

Traditional Fatigue Analysis

The initial strain can be considered to have the major effect on fatigue life. The average strain found from the first 100 cycles during the stress-controlled tests was therefore recorded as an independent variable. The fatigue test results are presented using the classical WÖHLER curve in Figure 8. The curves show the relationship between the fatigue life $N_f$ and initial strain, which is represented by a power fatigue law as shown in Equation (2).

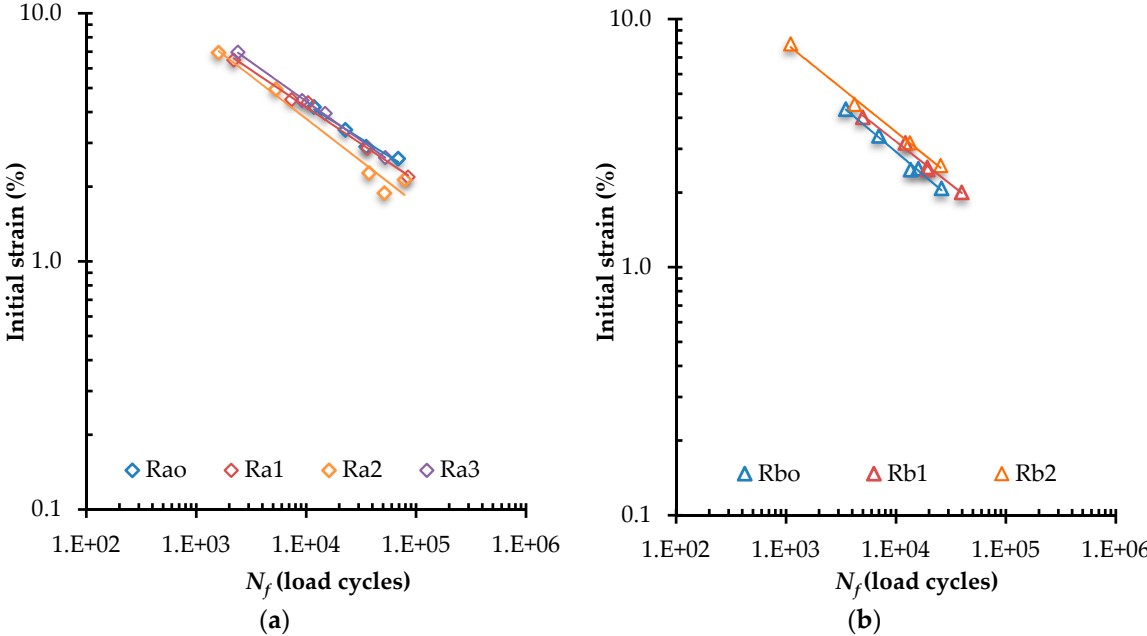

**Figure 8.** (**a**) Traditional fatigue curves for the virgin and Ra group binders; (**b**) Traditional fatigue curves for the virgin and Rb group binders.

The fatigue equations with their regression coefficient $R^2$ are presented in Table 3. The high $R^2$ values indicate the excellent relationship between fatigue lives and strain. The fatigue curves, in Figure 8a, demonstrate that the recovered binders Rao, Ra1 and Ra3 appeared to have comparable fatigue cracking resistance. Using the rejuvenator in the case of Ra3 has significantly increased the

fatigue life compared to Ra2 binder, which shares the same RA content with Ra3, but without using a rejuvenator. For the Rb group binders, Figure 8b, it can be seen that once again the difference in fatigue resistance of the different material is not significant. Having an RA percentage up to 60% does not seem to have a negative effect on the fatigue resistance of the materials.

**Table 3.** Fatigue equations and fatigue lives at 1% strain for the virgin and recovered binders.

| Material | Fatigue Equation | Fatigue Lives at 1% Strain | Regression Coefficient ($R^2$) |
|---|---|---|---|
| Rao | Nf = $1.72 \times 10^6 \varepsilon - 3.52$ | 1,720,000 | 0.97 |
| Ra1 | Nf = $1.21 \times 10^6 \varepsilon - 3.34$ | 1,210,000 | 0.99 |
| Ra2 | Nf = $4.28 \times 10^5 \varepsilon - 2.83$ | 428,000 | 0.97 |
| Ra3 | Nf = $1.10 \times 10^6 \varepsilon - 3.16$ | 1,100,000 | 0.99 |
| Rbo | Nf = $1.70 \times 10^5 \varepsilon - 2.84$ | 170,000 | 0.99 |
| Rb1 | Nf = $2.76 \times 10^5 \varepsilon - -2.84$ | 276,000 | 0.98 |
| Rb2 | Nf = $3.30 \times 10^5 \varepsilon - 2.79$ | 330,000 | 0.99 |

Cumulative Dissipated Energy

Cyclic loading of viscoelastic materials produces different paths for loading and unloading and therefore the formation of hysteresis loops. The dissipated energy per cycle is calculated as the area within the hysteresis loop using the following equation:

$$w_i \ = \ \pi \, \sigma_i \, \varepsilon_i \, \sin \delta_i \tag{4}$$

where $w_i$ = the dissipated energy at cycle *i*; $\sigma_i$, $\varepsilon_i$, $\delta_i$ = the stress amplitude, strain amplitude and phase angle at cycle i, respectively. Van Dijk and co-researchers applied the dissipated energy to characterise fatigue cracking in asphalt mixtures [24,29,30]. They demonstrated the relationship between accumulated dissipated energy ($W_{fat}$) at failure and the number of cycles ($N_{fat}$) to failure solely dependent on materials properties. The accumulated dissipated energy after *n* cycles is:

$$W_n = \sum_{i=0}^{n} w_i \tag{5}$$

The relationship between cumulative dissipated energy to the number of load cycles to failure can be represented by a power law relationship as follows:

$$W_{fat} = A. \, N_{fat}^z \tag{6}$$

where $W_{fat}$ = total dissipated energy until failure due to fatigue cracking, $N_{fat}$ = number of loading cycles to fatigue; and *A* and *z* = material constants.

The total dissipated energy until failure point (as defined by the DER approach) was computed using Equation (5) and the relationship between accumulative total energy and number of cycles at fatigue failure was established according to Equation (6). Figure 9a,b show the fatigue results based on total dissipated energy for the recovered binders from RA1 and RA2. In general, a material with a higher magnitude of total dissipated energy until failure results in a material with superior fatigue resistance. Thus, the fatigue results in Figure 9a,b agree with the traditional fatigue approach where all Ra and Rb group binders appeared to have comparable fatigue cracking resistance in comparison to the control binders.

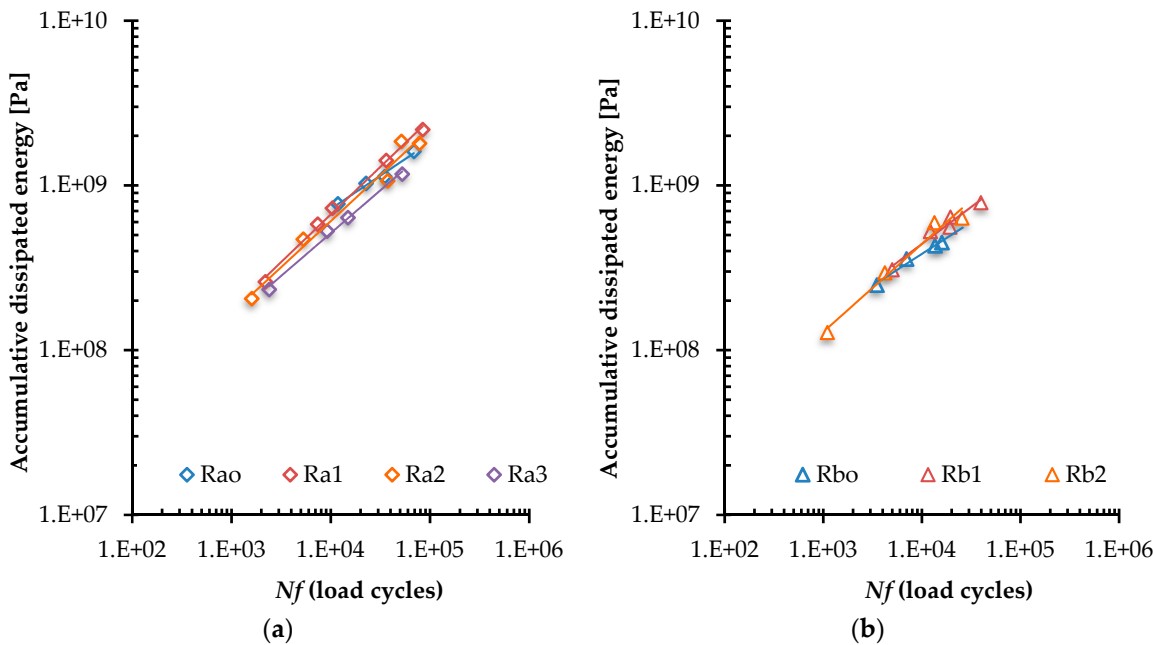

(a)　　(b)

**Figure 9.** (**a**) Total accumulative dissipated energy vs (*Nf*) for virgin binder and Ra group binders; (**b**) Total accumulative dissipated energy vs (*Nf*) for virgin binder and Rb group binders.

### 3.2. Asphalt Mixtures

Fatigue Performance

Indirect Tensile Fatigue Tests were performed to characterise the fatigue performance of the five asphalt recycled mixtures incorporating different amounts of RA, as well as the two control mixtures. The definition of the initial strain (1st or 100th load cycle) has an important influence on the final results due to significant changes in stiffness during the first few load cycles. The outcomes described in this paper involve the initial strain being defined at the 100th load cycle. Figures 10 and 11 show the developed fatigue curves for the tested asphalt mixtures. These curves show the relationship between the fatigue life $N_f$ and initial elastic strain, which can be expressed as a power fatigue law function as presented in Equation (3).

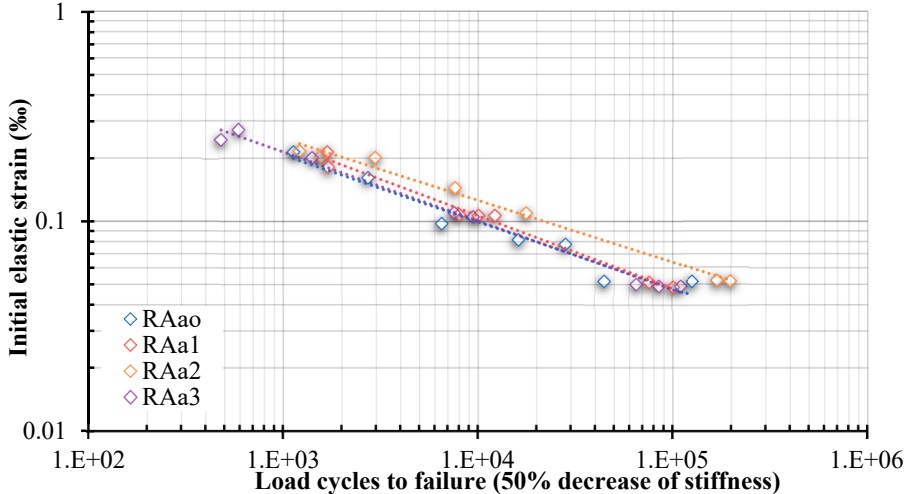

**Figure 10.** Fatigue curves for the tested asphalt mixtures SMA 8 S with "0% RA", "30% RA", "60% RA" and "60% RA + Rej".

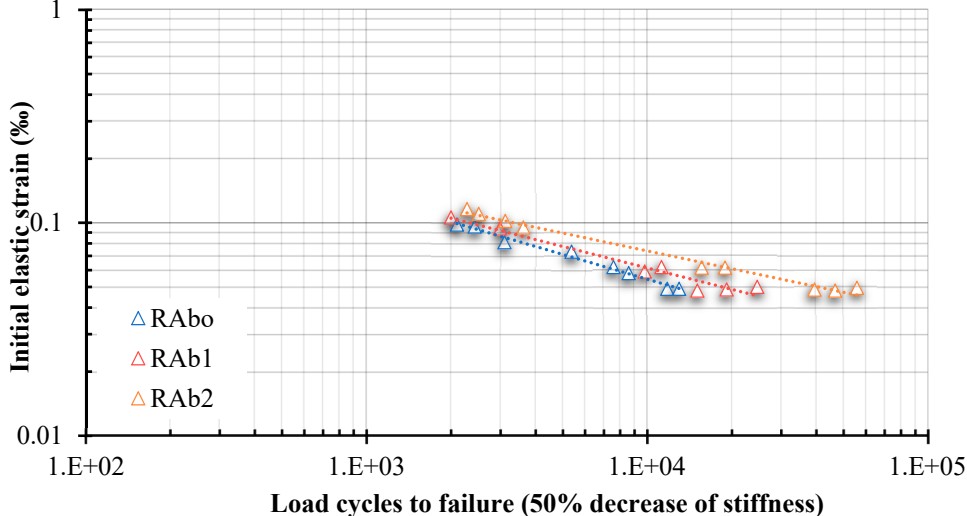

**Figure 11.** Fatigue curves for the tested asphalt mixtures AC 16 with "0% RA", "30% RA + Rej" and "60% RA + Rej".

The fatigue equations with their regression coefficient $R^2$ are presented in Table 4. The high $R^2$ values indicate the good relationship between fatigue lives and the resilient strains.

**Table 4.** Fatigue equations and fatigue lives at 100 µε for the control and recycled mixtures.

| Material | Fatigue Equation | Fatigue Lives at 100 µε (0.1 ‰ $\varepsilon_{ini}$) | Regression Coef. ($R^2$) |
|---|---|---|---|
| $RAa_o$ | $N_f = 1.137 \times 10^{-1} \varepsilon^{-3.067}$ | 10,275 | 0.95 |
| $RAa_1$ | $N_f = 4.627 \times 10^{-2} \varepsilon^{-2.786}$ | 13,189 | 0.99 |
| $RAa_2$ | $N_f = 1.500 \times 10^{-1} \varepsilon^{-3.509}$ | 21,509 | 0.99 |
| $RAa_3$ | $N_f = 9.651 \times 10^{-2} \varepsilon^{-3.012}$ | 10,653 | 0.99 |
| $RAb_o$ | $N_f = 1.303 \times 10^{-1} \varepsilon^{-2.463}$ | 2229 | 0.98 |
| $RAb_1$ | $N_f = 3.214 \times 10^{-1} \varepsilon^{-2.933}$ | 2664 | 0.97 |
| $RAb_2$ | $N_f = 1.2389\, \varepsilon^{-3.650}$ | 3603 | 0.99 |

The graphs provide visual confirmation that the stiffer asphalt material is, the higher the number of load cycles until failure. The fatigue resistance therefore increases with an increasing amount of RA, especially at small initial elastic strains. The higher the initial elastic strain, the smaller the influence of the amount of RA.

The fatigue functions in Figures 10 and 11 and the fatigue parameters for the asphalt mixtures in Table 4 demonstrate good fatigue performance of the recycled asphalt mixtures, particularly the proprietary 60% RA asphalt mixtures. The fatigue results also confirm the superior fatigue resistance of SMA manufactured with PmB binder in comparison to the AC mixes that were manufactured with neat bitumen.

The number of load cycles to fatigue failure in an actual pavement cannot be estimated directly from the outcomes of laboratory fatigue tests. Different factors such as duration of rest periods between traffic loading (and rest periods), lateral traffic wander and crack propagation all influence the number of load cycles to fatigue failure in the real pavement. These fatigue cycles to failure will always be higher than the number of load cycles to fatigue determined in the laboratory.

## 4. Conclusions

The rheological and fatigue-related properties of binders recovered from different asphalt mixtures were assessed using different analysis approaches. Mechanical tests were also performed on these plant-produced mixtures. The binders were recovered from asphalt mixtures that had been

manufactured in asphalt plants using different sources and amounts of RA in contents of up to 60% of RA, with and without rejuvenating additives. The rheological and fatigue-related properties of the binders were assessed using DMA, traditional fatigue analysis and cumulative dissipated energy. The mechanical tests on asphalt mixtures were assessed by ITFT.

The main conclusions that could be drawn based on laboratory investigations in this paper are:

1.  In the case of the moderately aged RA, Ra binders group, the recovered binders, even without using rejuvenators, showed similar performance or even slightly improved performance to that of the control binder. However, when the rejuvenator was added, the fatigue-related performance properties were significantly improved.

2.  In the case of the stiff RA, Rb binders group, the use of a rejuvenator was essential even with 30% of RA with the fatigue-related properties not been negatively affected by using up to 60% of RA.

3.  Regarding the recycled mixtures, all mixtures show adequate resistance to fatigue. Up to a medium-high (30–60%) percentage of RA, the resistance to fatigue increases with increasing RA quantity.

4.  Results of this investigation highlight the potential to maximise the RA content in hot mix asphalt for surface courses. The results have shown that stiffer reclaimed asphalt binders have the potential to produce comparable performance to virgin binders, with satisfactory results when up to 60% RA is used. Despite the promising results for binders recovered from asphalt plant mixtures, there is still a gap to be investigated in linking the results of the binders with the asphalt mixtures.

**Author Contributions:** Conceptualization, A.S., G.M.P. and D.L.P.; Methodology, A.S., G.M.P., D.L.P., A.J.d.B.C. and G.A.; Formal Analysis, A.S. and G.M.P.; Investigation, G.M.P.; Writing—Original Draft Preparation, A.S. and G.M.P.; Writing—Review and Editing, G.A.; Supervision, G.A. and D.L.P.

**Funding:** The second author is supported by Conselho Nacional de Desenvolvimento Científico e Tecnológico, CNPq—Brazil (Grant No. 200279/2014-9).

**Conflicts of Interest:** The authors declare no conflict of interest.

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
