# Peer review of "Binder and Mixture Fatigue Performance of Plant-Produced Road Surface Course Asphalt Mixtures with High Contents of Reclaimed Asphalt"

_sustainability, doi:10.3390/su11133752_

Round 1
Reviewer 1 Report
Overall, the topic is interesting but since there are so many studies on the characterization of RA materials that the novelty and contribution of the work is difficult to identify.
Ln38: ‘factors’ should be ‘factor’
Ln47: ‘economics’ should be ‘economic’
Ln92: Please consider including the service life time of the sections that are used in this study, if available.
Ln116: The statement regarding the similarity of the physical properties is not supported by the data; except from the Fraas point, the other properties are not similar.
Ln144: It would be interesting to include information on the mix production such as type of plant, preheating of RAP material, etc.
Ln215: Did the authors attempt to relate the fatigue properties of the recovered binders and the mixtures? Or the binder stiffness and fatigue life? Please consider adding a section with this information.
Ln303: The results from past laboratory studies (including the current one) show that fatigue life increase with the addition of RA and ageing in general. However, this observation contradicts field observations (where fatigue cracking is higher for RA and highly-aged pavements) and raises questions about the suitability of such criteria. Could the authors comments on the suitability of fatigue tests to characterize RA materials wrt their sensitivity to fatigue cracking?
Author Response
Thanks very much for the review. Please, see below our responses in blue.
Ln38: ‘factors’ should be ‘factor’: fixed
Ln47: ‘economics’ should be ‘economic’ fixed
Ln92: Please consider including the service life time of the sections that are used in this study, if available. Sorry, we do not have the information.
Ln116: The statement regarding the similarity of the physical properties is not supported by the data; except from the Fraas point, the other properties are not similar. Yes, we agree. The statement has been deleted.
Ln144: It would be interesting to include information on the mix production such as type of plant, preheating of RAP material, etc.
It is batch-type with a parallel drum and the full details of mix production can be found in this reference "Wellner, F.; Canon Falla, G.; Milow, R.; Blasl, A.; Di Mino, G.; Di Liberto, C. M.; Noto, S.; Lo Presti, D.; Jimenez del Barco Carrion, A.; Airey, G. “AllBack2Pave: High-content RA warm asphalt mixture design,” CEDR Deliverable 2015."
Ln215: Did the authors attempt to relate the fatigue properties of the recovered binders and the mixtures? Or the binder stiffness and fatigue life? Please consider adding a section with this information. T
The fatigue properties of different binders were very similar so it is difficult to draw a robust conclusion about the correlation between binders and mixtures.
Ln303: The results from past laboratory studies (including the current one) show that fatigue life increase with the addition of RA and ageing in general. However, this observation contradicts field observations (where fatigue cracking is higher for RA and highly-aged pavements) and raises questions about the suitability of such criteria. Could the authors comments on the suitability of fatigue tests to characterize RA materials wrt their sensitivity to fatigue cracking?
Yes, it is acknowledged that stiffer materials tend to have superior fatigue life especially when tested under 'stress-controlled mode'. Some findings in the literature showed that the fatigue life of stiffer materials is normally longer than the fatigue life of softer materials when tested under controlled-stress loading. The results can change when materials tested under strain-controlled condition. The rational behind fatigue properties is rather complicated and involve many variables, test conditions, materials, the defined failure point. The use of 'initial strain' in this paper as fatigue dependent criteria rather than the stress can somehow reduce the biased effect of stiffer materials.
Reviewer 2 Report
GENERAL COMMENTS The paper presents an interesting experimental study involving mixtures prepared at real scale and aimed at demonstrating the feasibility of producing hot mix asphalts with high content of RAP. The introduction is well organized and the relevant literature is cited even if improvements are possible (see specific comments below). The objective of the paper is well stated. Overall, the methodology is well presented. Based on the information reported, a colleague should be able to reproduce the experimentation, even if further information can be added for a more comprehensive picture (see specific comments below). Some criticisms could be detected in the experimental design (excessive heterogeneity) due to the high number of selected variables (2 RAP sources, with/without rejuvenators, 2 types of mixtures, different recycling rates, 2 bitumens, etc.). However, this provides a wide picture of available options in hot mix in-plant recycling. Gnerally, the experimental findings are properly justified and commented even if some flaws can be detected in the analysis of the results (see specific comments below). Conclusions are partly affected by the abovementioned flaws. SPECIFIC COMMENTS Lines 62-66: an interesting Italian research experience having the same objective and specifically addressed to porous asphalt mixtures can be also cited. See for example: “Laboratory study to evaluate the influence of reclaimed asphalt content on performance of recycled porous asphalt” or “Improved durability of recycled porous asphalt” or “Use of reclaimed asphalt in porous asphalt mixtures: laboratory and field evaluations” Reference 12 is on cold recycling. It does not seem appropriate for this paper. How are the Authors sure that the difference in penetration of the two recovered binders comes from a different aging instead of different original properties of unaged bitumens? Based on Journal policy, the Authors must decide if citing or not he Standards in the references list. In the current version, only EN 12697-4 is cited and many others no. Please be coherent. Line 135: what does “TL-Asphalt StB 07 (2013)” mean? Is a citation in the reference list necessary? Please comment about the fact that a different granulometric distribution among RAa mixtures (as well as among RAb mixtures) could decisively affect the final performances of the mixes, hiding the influence of RAP as well as of the other variables. Only information about the total bitumen content of the tested mixtures is provided without specifying the amount of bitumen coming from the RAP and the amount of added virgin bitumen. It is suggested to introduce the bitumen content of the two RAP sources in section 2.1 (Table 1?). Line 150: what does “(ANAS, 2010)” mean? Is a citation in the reference list necessary? Please comment about the very low void content of the dense graded mixture. And what about the production temperature higher than that of the SMA prepared with a PmB modified bitumen? Lines 164-165: avoid commercialisms How many test repetitions for the binder tests? Why did the Authors perform also frequency sweeps? How is it related to the fatigue characterization? What about the strain amplitude for the frequency sweeps? Was the LVE limit determined through amplitude sweeps? Can you add brief information? What about specimens’ dimensions for ITFT tests? Line 201: are the Authors referring to the initial strain? At what loading cycle? And what about the corresponding stresses? Line 203: initial = what loading cycle? Section 3.1.1: can the Authors add the Black diagrams? This would also allow to establish the applicability of the TTSP Lines 229-231: the effect at low temperature due to the use of the rejuvenator seems different for the two types of mixtures. Please elaborate Table 3: please comment about the noticeable difference in fatigue behavior of the two types of mixes Line 289: what does it mean “around”? Please be more specific Lines 302-305: this is true if the material works at controlled stress in the field. It is the opposite if the material works at controlled strain. The real behavior in the field depends on the mechanical properties of the whole pavement, but in the case of a wearing course is usually more “strain-controlled” than “stress-controlled”. In fact, at lines 227-228 the Authors stated that an increase in stiffness could negatively affect the fatigue-related properties. Could the Authors elaborate about this point? Lines 332-335: this does not seem in line with the results presented before. Lines 336-340: please be more cautious
Author Response
SPECIFIC COMMENTS
Lines 62-66: an interesting Italian research experience having the same objective and specifically addressed to porous asphalt mixtures can be also cited. See for example: “Laboratory study to evaluate the influence of reclaimed asphalt content on performance of recycled porous asphalt” or “Improved durability of recycled porous asphalt” or “Use of reclaimed asphalt in porous asphalt mixtures: laboratory and field evaluations”
- Thank you for suggesting these interesting studies. They have been added as references.
Reference 12 is on cold recycling. It does not seem appropriate for this paper.
- Although reference 12 is for cold recycling, the authors believe that it is important to mention this research effort dedicated to improve practices with recycled asphalt
How are the Authors sure that the difference in penetration of the two recovered binders comes from a different aging instead of different original properties of unaged bitumens?
- The authors agree with the uncertainty about the age of the RA and therefore have modified the text accordingly.
Based on Journal policy, the Authors must decide if citing or not he Standards in the references list. In the current version, only EN 12697-4 is cited and many others no. Please be coherent.
- Thank you, the standard citation has been removed.
Line 135: what does “TL-Asphalt StB 07 (2013)” mean? Is a citation in the reference list necessary?
- TL Asphalt StB 07 is the German Standard for grading limits and therefore it is not necessary to add it in the reference list, but it has been explained in the manuscript
Please comment about the fact that a different granulometric distribution among RAa mixtures (as well as among RAb mixtures) could decisively affect the final performances of the mixes, hiding the influence of RAP as well as of the other variables.
- It is understood the inherit heterogeneity of the RA affect the overall performance of the final mix including stiffness, fatigue resistance, rutting and moisture damage. In this work, the amount and gradation of virgin aggregate were altered so that the final gradation of the mixes with RA comply with the gradation limits. Also, the experiment was designed with large difference in RAP content, i.e. 30% difference in RAP between the different mixes to identify clearly the RAP variable.
Only information about the total bitumen content of the tested mixtures is provided without specifying the amount of bitumen coming from the RAP and the amount of added virgin bitumen. It is suggested to introduce the bitumen content of the two RAP sources in section 2.1 (Table 1?).
- The binder content of each RA source has been added in line 97
Line 150: what does “(ANAS, 2010)” mean? Is a citation in the reference list necessary?
- It refers to “Capitolato speciale d’appalto – Norme Tecniche. Azienda Nazionale Autonoma delle Strade, Italy”. The text has been modified to clarify
Please comment about the very low void content of the dense graded mixture.
- The target air voids were 4%-8%, however, the low air voids are attributed to the relative increase in the effective binder content and the use of rejuvenators.
And what about the production temperature higher than that of the SMA prepared with a PmB modified bitumen?
- The maximum temperature to manufacture SMA and AC mixtures was the same: 170ºC. This temperature is high in the case of the SMA because of the PmB, and because of the high RA stiffness in the case of the AC.
Lines 164-165: avoid commercialisms
- Brands have been removed
How many test repetitions for the binder tests? Why did the Authors perform also frequency sweeps? How is it related to the fatigue characterization? What about the strain amplitude for the frequency sweeps? Was the LVE limit determined through amplitude sweeps? Can you add brief information? What about specimens’ dimensions for ITFT tests?
- Three repetitions were done for each test and the average were reported. All this information has been added in the manuscript. Frequency sweeps were performed to compare the stiffness of the binders, see the effect of the rejuvenators and relate to fatigue performance
Line 201: are the Authors referring to the initial strain? At what loading cycle? And what about the corresponding stresses? Line 203: initial = what loading cycle?
- The initial strain being defined at the 100th load cycle for the asphalt mixes and the average strain during the first 100 cycles for the binders. They are stress-controlled tests. It is explained in Section 3.1.2.1 and Section 3.2.1.
Section 3.1.1: can the Authors add the Black diagrams? This would also allow to establish the applicability of the TTSP.
- They are added now.
Lines 229-231: the effect at low temperature due to the use of the rejuvenator seems different for the two types of mixtures. Please elaborate
- RA and virgin binder used in each type of mixture were different, so the direct comparison of the effect of the rejuvenator is not immediate.
Table 3: please comment about the noticeable difference in fatigue behavior of the two types of mixes
- The fatigue results confirm the superior fatigue resistance of SMA manufactured with PmB binder in comparison to the AC mixes that were manufactured with neat bitumen. This has been added.
Line 289: what does it mean “around”? Please be more specific
- More details have been added
302-305: this is true if the material works at controlled stress in the field. It is the opposite if the material works at controlled strain. The real behavior in the field depends on the mechanical properties of the whole pavement, but in the case of a wearing course is usually more “strain-controlled” than “stress-controlled”. In fact, at lines 227-228 the Authors stated that an increase in stiffness could negatively affect the fatigue-related properties. Could the Authors elaborate about this point?
- Stiffer materials tend to sustain less strains at failure and develop larger thermal stresses which all contribute negatively to the fatigue-related properties. We agree the fatigue is very complex area and many variables involve in determine the response of mixes, i.e. test conditions, stiffness, fatigue failure criteria. Yes, some findings in the literature showed that the fatigue life of stiffer materials is normally longer than the fatigue life of softer materials when tested under controlled-stress loading. The effect of stiffness on fatigue life can be reduced when the fatigue curves are plotted as a function of the initial strain and not as a function of stress level.
Lines 332-335: this does not seem in line with the results presented before.
- This statement has been removed
Lines 336-340: please be more cautious
- Conclusions have been rephrased